# Highly Sensitive Detection of PQS Quorum Sensing in Pseudomonas Aeruginosa Using Screen-Printed Electrodes Modified with Nanomaterials

**DOI:** 10.3390/bios12080638

**Published:** 2022-08-13

**Authors:** Denisa Capatina, Teodora Lupoi, Bogdan Feier, Diana Olah, Cecilia Cristea, Radu Oprean

**Affiliations:** 1Department of Analytical Chemistry, Faculty of Pharmacy, “Iuliu Hatieganu” University of Medicine and Pharmacy, 4 Pasteur Street, 400349 Cluj-Napoca, Romania; 2Department of Infectious Diseases and Preventive Medicine, Faculty of Veterinary Medicine, University of Agricultural Sciences and Veterinary Medicine Cluj-Napoca, Calea Manastur 3-5, 400372 Cluj-Napoca, Romania

**Keywords:** PQS, quorum sensing, *P. aeruginosa*, electrochemical sensor, electrochemical detection, screen-printed electrode, microbiological cultures analysis, urine sample analysis

## Abstract

The rapid diagnosis of *Pseudomonas aeruginosa* infection is very important because this bacterium is one of the main sources of healthcare-associated infections. Pseudomonas quinolone signal (PQS) is a specific molecule for quorum sensing (QS) in *P. aeruginosa*, a form of cell-to-cell bacterial communication and its levels can allow the determination of the bacterial population. In this study, the development of the first electrochemical detection of PQS using screen-printed electrodes modified with carbon nanotubes (CNT-SPE) is reported. The electrochemical fingerprint of PQS was determined using different electrode materials and screen-printed electrodes modified with different nanomaterials. The optimization of the method in terms of electrolyte, pH, and electrochemical technique was achieved. The quantification of PQS was performed using one of the anodic peaks in the electrochemical fingerprint of the PQS on the CNT-SPE. The sensor exhibited a linear range from 0.1 to 15 µM, with a limit of detection of 50 nM. The sensor allowed the selective detection of PQS, with low interference from other QS molecules. The sensor was successfully applied to analysis of real samples (spiked urine and human serum samples, spiked microbiological growth media, and microbiological cultures).

## 1. Introduction

*P. aeruginosa* is a rod-shaped Gram-negative bacterium that is ubiquitous in the healthcare environments, such as nursing homes, and hospitals. There are many types of infections caused in humans by *P. aeruginosa*, affecting the lungs, urinary tract, ears, eyes or skin and they usually occur in immunosuppressed individuals. The high ability to develop antimicrobial resistance (AMR) and form biofilms makes *P. aeruginosa* a challenging bacterium, being included on the World Health Organization’s short list of antimicrobial-resistant critical, high-priority pathogens—the ESKAPE group (*Enterococcus faecium*, *Staphylococcus aureus*, *Klebsiella pneumoniae*, *Acinetobacter baumannii*, *P. aeruginosa*, and *Enterobacter* spp.). For all these reasons, the rapid diagnosis of *P. aeruginosa* infection is very important [1,2,3].

*P. aeruginosa* presents complex molecular mechanisms in order to assure its survival during the pathogenesis and in ever-changing conditions of the environment. One of these mechanisms is quorum sensing (QS), which is a form of cell-to-cell communication between bacteria, which involves multiple interconnected signal transduction pathways. The QS enables the individual bacteria to collect information about the cell density and the environmental conditions of the bacterial population, leading to a collective behavior [1,2,3,4].

The bacteria communicate with each other through autoinducers (AIs), which are small signaling molecules produced and excreted by bacteria in the extracellular medium. The number of AIs is proportional to the population density, so the detection of QS molecules can be a useful tool for monitoring the bacterial growth [4,5].

*P. aeruginosa* produces different AIs, such as N-3-oxo-dodecanoyl L-homoserine lactone (3-O-C_12_-HSL), N-butyryl L-homoserine lactone (C_4_-HSL), 2-heptyl 3-hydroxy-4-quinolone, known as *Pseudomonas* quinolone signal (PQS), its precursor 2-heptyl-4-hydroxyquinoline (HHQ), and 2-(2-hydroxyphenyl) thiazole-4-carbaldehyde (IQS). When a threshold concentration of these QS molecules is detected by the bacteria, they modify their gene expression leading to the activation of certain metabolic processes, the secretion of virulence factors, and/or the biofilm formation [4,6].

PQS is a molecule secreted only by *P. aeruginosa* and is the most representative QS molecule for this bacterium. It acts as a strong agonist of the transcriptional regulator PqsR and induces the expression of various virulence factors, such as elastase, pyocyanin (PYO), lectins, and hydrogen cyanide. It also plays a key role in biofilm formation through the release of extracellular DNA, a major component of the biofilm matrix [1,7,8].

PQS has been shown to have a positive association with the disease status and is also associated with the onset of pulmonary exacerbation in *P. aeruginosa* infections. Therefore, quantification of this molecule in biological fluids could provide important information to characterize the infection and help physicians to establish an effective treatment [1,7,9].

PQS is found at very low concentrations in biological fluids and culture media (nM range in sputum, urine or plasma and low µM range in biofilms and culture media) [1,6,10]. Therefore, very sensitive and specific methods are required for its detection and quantification in these complex matrices.

Conventional detection methods for PQS include the use of HPLC and MS techniques, which can quantify PQS in biological fluids in the low nM range [7,8,10]. Although these methods are very sensitive and selective, they require complex and long work protocols, expensive equipment, expensive and polluting solvents, and do not allow decentralized analyses [1,8]. The enzyme-linked immunosorbent assay has also been shown to be a suitable approach for the sensitive detection of PQS in culture media, but it has certain limitations under non-laboratory or field conditions, due to the long analysis time and the sophisticated equipment required [1,9]. Electrochemical methods can overcome these limitations, as they are simple and sensitive approaches for detecting PQS even in complex matrices [1]. Several electrochemical sensors have been described for the detection and quantification of these molecules in biological fluids and culture media [11,12,13,14,15,16]. All these sensors used BDDE or GCE as the working electrode for the PQS detection, but these electrodes are not very easy to use in decentralized analyses.

Screen-printed electrodes (SPEs), using various electrode formats and materials, have been successfully applied for the field detection of a wide range of analytes in different biological samples. Their extensive use is due to the fact that they are mass produced at low cost, disposable after one use (eliminating thus the problems associated with the electrode fouling), suitable for miniaturization and automation procedures, they are easy to use, require small volumes of the samples, and assure short analysis time. While the phenomenon of resistance is increasing and *P. aeruginosa* also possesses the ability to form a biofilm, the need for point-of-care detection is essential. A rapid, sensitive, and cheap detection that can be achieved through SPEs gives more time for the health professionals to establish a treatment scheme and increase the chances of survival of *P. aeruginosa*-infected patients. The progress in the printing technique has allowed the modification of the electrode surface with many nanomaterials or bio-elements and the production of stable, reproducible sensors [17,18].

Nanomaterials (graphene (GPH), carbon nanotubes (CNT), metallic nanoparticles (NPs)) are very popular for the fabrication of electrochemical platforms, due to their properties, such as good conductivity, large surface area, biocompatibility, ease of modification, and functionalization [19].

In this study, a fast, simple, and sensitive method for the detection of PQS was developed, employing the peak obtained by its electrochemical oxidation using commercially available SPE modified with CNTs (CNT-SPE). Other electrode materials, such as glassy carbon electrode (GCE), carbon paste electrode (CPE), and several commercially available SPEs based on boron-doped diamond (BDD-SPE), platinum (Pt-SPE), carbon (C-SPE), carbon modified with gold NPs (AuNPs), carbon-based nanomaterials (ordered mesoporous carbon (OMC-SPE) and GPH (GPH-SPE)) and gold NPs (GNP-SPE), were tested, with the highest and most reproducible signal obtained for CNT-SPE. The optimal conditions in terms of electrolyte, pH, and electrochemical technique were determined. The CNT-SPE allowed the sensitive and selective detection of PQS, and the method was successfully applied to PQS detection from real samples. To the best of our knowledge, this is the first time that the electrooxidation behavior of PQS was evaluated using SPEs, the developed sensor using CNT-SPE presenting improved analytical performance and being a promising tool for POC detection of *P. aeruginosa*.

## 2. Materials and Methods

### 2.1. Materials 

All the reactives had analytical purity and were used without further treatment. H_2_SO_4_, Na_2_HPO_4_, NaH_2_PO_4_, HAuCl_4_, NaOH, PQS, C_4_-HSL, 3-O-C_12_-HSL, uric acid (UA), glucose (GLU), PYO, acetaminophen (APAP), and human serum were purchased from Sigma-Aldrich (St. Louis, MO, USA); ethanol from Titolchimica (Pontecchio polesine, Italy); glacial acetic acid from Merck (Rahway, NJ, USA); gentamicin sulphate (GEN) was purchased from BioWorld (St. Louis Park, MN, USA); ceftazidime pentahydrate (CEF) was received from Antibiotice SA Iasi (Iasi, Romania). The microbiological experiments used nutrient broth (NB) (Cooked Meat Medium, Oxoid, UK). All the solutions were prepared in ultrapure water. The urine sample was collected from a healthy volunteer.

### 2.2. Instruments 

The electrochemical measurements were performed on a Autolab PGSTAT 302N (Metrohm Autolab, Utrecht, The Netherlands) equipped with the associated Nova 1.10.4 software. The GCE and CPE were used as working electrodes in a conventional three-electrode cell with a Pt wire as the counter electrode and Ag/AgCl KCl 3 M as the reference electrode, obtained from BAS Inc. (West Lafayette, IN, USA). The GCE had a geometric surface area of approximately 0.12 cm^2^ and was polished with an alumina suspension and polishing cloth before each analysis. The SPEs were purchased from Metrohm Dropsens (Oviedo, Spain) and they had different working electrode materials: graphite (C-SPE), boron-doped diamond (BDD-SPE), platinum (Pt-SPE), gold NPs (GNP-SPE), GPH (GPH-SPE), ordered mesoporos carbon (OMC-SPE), and CNTs (CNT-SPE).

### 2.3. Methods

#### 2.3.1. Electrochemical Methods 

In order to choose the best electrode material and the most suitable electrolyte for the detection of PQS, cyclic voltammetry (CV) and differential pulse voltammetry (DPV) were employed. For CV measurements, the potential window (PW) was from −1 to 1.3 V, the scan rate (SR) 0.05 V s^−1^, the step potential (SP) 0.00244 V, and the interval time 0.048828 s. For DPV the PW was −0.1 to 1.3 V, the SP 0.005 V, the modulation amplitude (MA) 0.1 V, and modulation time (MT) 0.025 s.

To choose the appropriate electrochemical technique for PQS detection, two voltametric methods were studied: square wave voltammetry (SWV) and DPV. For SWV, the PW was from 0 to 1.3 V, SP 0.005 V, amplitude 0.025 V, and frequency of 10 Hz. The DPV parameters remained the same as mentioned before.

For the quantification of PQS, DPV was used with the following parameters: PW from −0.1 to 1.3 V, SP 0.005 V, MA 0.1 V, MT 0.025 s, interval time 0.5 s, and an SR of 0.01 V s^−1^.

The modification of the C-SPE with AuNPs was performed by CV, as previously reported [20]. Two solutions of 1.5 and 5 mM of HAuCl_4_ prepared in 0.5 M H_2_SO_4_ were used for the electrodeposition of AuNPs, by performing 35 CV cycles in a PW from −0.2 to 1.2 V, with an SR of 0.1 V s^−1^.

#### 2.3.2. Quantification Method

The analyte solution was first prepared as a stock solution in ethanol and kept in the freezer in order to avoid degradation. The desired concentrations were obtained by diluting the stock solution with ultrapure water. To detect and quantify the target molecule, the signal given during the oxidation process was recorded. The current intensity of the first peak given by the molecule was measured as it is the most significant peak and the first one to appear at low concentrations. The peak was measured in DPV from the baseline around 0.2 V potential. For the quantification of the molecule in real samples, a shift of the oxidizing potential was observed at 0.45 V for NB and 0.35 V for urine.

#### 2.3.3. Interference Studies

The selectivity of the method was tested taking into account the most common components of the analyzed samples. From the class of drugs used in the therapy of *P. aeruginosa* infections, GEN, CEF, and APAP were tested. The second class whose interference was verified was the AI molecules secreted by *P. aeruginosa* with homoserine lactone structure, C_4_-HSL and 3-O-C_12_-HSL. UA and GLU were tested because of their presence in serum and urine. The last interferent was PYO, a major electrochemically active secondary metabolite of *P. aeruginosa*. The influence of the interferents was tested by measuring the signal of a PQS solution in the presence of each of the before mentioned molecules. In the solution of the mixture, the proportion between PQS and GEN, CEF, C_4_-HSL, 3-O-C_12_-HSL, and PYO was 1:1, for APAP, UA, and GLU the proportion was 1:100 due to their presence in a higher concentration compared to PQS in the biological samples. The signal of a 5 µM solution of GEN, CEF, C_4_-HSL, 3-O-C_12_-HSL, PYO, and 500 µM of APAP, UA, and GLU was also recorded. The percentage of PQS in the mixture calculated from the signal of PQS only displayed the degree of interference.

#### 2.3.4. Real Samples Analysis

The real samples studied were NB, urine, and serum. Both spiked and real *P. aeruginosa* cultures were analyzed for NB. PQS was added to the collected urine, serum, and NB samples and then subjected to a simple treatment based on dilution of the samples in a 1:10 ratio with the electrolyte H_2_SO_4_ 0.5 M, the final concentration of the analyte being 1 µM. To detect the influence of the matrix in these complex samples on the PQS signal, the standard addition method was used: a small volume of a standard PQS solution was added three times to the diluted samples increasing the concentration of the solution with 1, 2, and 3 µM, respectively. The recovery was calculated by extrapolating the calibration plot to zero.

For *P. aeruginosa* culture analysis, a standard strain (ATCC 27853) and a clinical isolate were submitted to aerobic incubation for 72 h at 37 °C in 10 NB. Following the established protocol, the cultures were sampled, inactivated, and tested after 16, 24, 48, and 72 h of incubation. Before analysis, samples were simply diluted with H_2_SO_4_ 0.5 M electrolyte in a ratio of 1:10. The DPV signal around 0.4 V was measured to quantify the PQS. In parallel with the electrochemical analysis, the number of colonies was counted.

All procedures and analyses were performed with the approval of the Ethics Committee of “Iuliu Hatieganu” University of Medicine and Pharmacy Cluj-Napoca, Romania (147/30 March 2020), and in accordance with the principles of the Declaration of Helsinki.

#### 2.3.5. Estimation of the Number of Microorganisms

A suspension of 9 mL of saline and 1 mL of the initial bacterial suspension was prepared at each test point, relative to the dilution obtained by using H_2_SO_4_ 0.5 M for inactivation. Because of the high turbidity of the suspension, decimal dilutions up to 10^−11^ were made. Then, 9 mL of saline were distributed into a series of sterile culture tubes: 1 mL of the bacterial suspension was added to the first tube, resulting in a 1/10 dilution; then, 1 mL of the 1/10 dilution was added to another tube with another pipette, resulting in a 1/100 dilution, and so on. From each dilution, 1 mL of the solution was poured into a Petri dish. Then, 12–15 cm^3^ of nutrient agar (Nutrient Agar, Tulip Diagnostics, Verna, India), which had been melted and cooled to 45 °C, was added to the plates. Homogenization of the plate contents was achieved by horizontal rotation. The plates were then allowed to solidify for 30 min and then incubated at 37 °C for 24 h. After incubation, the colonies that had developed both on the surface and in the depth of the agar were counted. Only the plates with a colony density that allowed counting and maintained the dilution ratio were counted. The number of bacteria that grew at 37 °C in 24 h was expressed in colony forming units (CFU) and calculated using Equation (1):(1)$$TNG\;\left(CFU\;mL^{-1}\right)\;=\;\frac{\Sigma \left(n\;d\right)}{N}$$
where:

n = the number of colonies developed after incubation in a Petri dish;

d = the dilution with which the respective plate was inoculated;

N = the number of Petri dishes considered.

## 3. Results and Discussion

### 3.1. Electrochemical Detection of PQS

#### 3.1.1. Electrochemical Behavior on Different Electrode Materials

The electrochemical behavior of a 50 µM PQS solution was studied on different electrode materials by CV in a wide potential window, from −1 to + 1.3 V (Figure 1).

GCE, CPE, and several commercially available SPEs (C-SPE and C-SPE modified with AuNPs, BDD-SPE, Pt-SPE, GNP-SPE, OMC-SPE, CNT-SPE, and GPH-SPE) were tested to select the most suitable electrode surface for PQS detection. On all SPEs, except Pt-SPE, the recorded voltamograms showed four peaks in oxidation, with an irreversible behavior. In all cases, the highest peak was around 0.2 V. On Pt-SPE, a single oxidation peak could be observed at 0.58 V (Figure 1C), while on GCE and CPE two peaks were visible around 0.6 and 1.2 V, respectively (Figure 1J,K). PQS shows a reduction peak only on BDD-SPE at −0.3 V (Figure 1B). All the other electrode materials had no significant peak in reduction specific for PQS (Table 1). In the case of GNP-SPE and C-SPE modified with AuNPs, a reduction peak around 0.5 V can be observed, but this corresponds to the reduction of gold on the surface of the electrode, being also observed in the blank (Figure 1D–F). On BDD-SPE, the first oxidation peak shifts to a more positive potential (0.430 V) and has the smallest value of all electrodes tested (Figure 1B).

A widely used approach for increasing the sensitivity of the detection method is to modify the electrode surface with nanomaterials. We have shown in a previous study that the modification of C-SPE with AuNPs by electrodeposition of a HAuCl_4_ solution leads to an increase in the electroactive surface area and electrical conductivity, resulting in a significant improvement in electron transfer [20]. This is also confirmed in this study, as modification of the carbon surface with AuNPs by this method resulted in higher peaks for PQS compared to C-SPE. A disadvantage of this modified electrode surface is the appearance of additional peaks at potentials above 0.8 V because of gold oxidation, which prevents the detection of the peaks corresponding to the PQS molecule at higher potential values (Figure 1D,E). The increase in HAuCl_4_ concentration leads to an increase in PQS peaks, but at the same time to an increase in the peak corresponding to the oxidation of gold. The facilitation of electron transfer by surface modification with AuNPs was not as visible in the case of the commercially available GNP-SPE. The PQS peaks were smaller compared to those obtained with C-SPE/AuNPs and even smaller than those obtained with C-SPE (Figure 1F).

The best results were obtained on SPEs modified with carbon-based nanomaterials (Table 1). Of these, the highest peaks were obtained on the OMC-SPE (Figure 1G), followed by the CNT-SPE (Figure 1H) and the GPH-SPE (Figure 1I). These results were also confirmed by DPV (Figure 2). A drawback of the OMC-SPE is the appearance of a peak in the analysis of the blank (H_2_SO_4_ 0.5 M), which overlaps with the peak corresponding to PQS (Figure 2, blue). Although on CNT-SPE the peak corresponding to PQS is smaller, it does not show any peak in the blank (Figure 2, red).

To see if there is a correlation between the peak height and the molecule concentration in solution when analyzing PQS on OMC-SPE, different concentrations of PQS in 0.5 H_2_SO_4_ were analyzed. The presence of the peak in the blank negatively influences the PQS analysis, especially at low concentrations. The separation of the two peaks is very weak and therefore the reading of the peak corresponding to the PQS is difficult (data presented in Appendix A).

Although the recorded signal is higher on the OMC-SPE, we chose the CNT-SPE as the working surface because of the limitations of OMC-SPE for PQS detection.

#### 3.1.2. Influence of Electrolyte pH on the Detection of PQS

The DPV signal of 50 µM PQS was studied on CNT-SPE in different electrolytes and at different pH values: H_2_SO_4_ 0.01 M, H_2_SO_4_ 0.1 M, H_2_SO_4_ 0.5 M, H_2_SO_4_ 1 M, acetate buffer 0.1 M, pH 4 and pH 5, PB 0.02 M pH 2, 3, and 4, PBS 0.02 M, pH 6, 7, 8, and 9, and NaOH 0.5 M (Figure 3).

The electrolyte and pH influences both the peak potential and the peak amplitude (Table 2). Four peaks can be observed at acidic pH, generally around 0.2, 0.5, 0.8, and 1 V. Exceptions are the analyses in 0.01 M H_2_SO_4_ when only one peak is observed at 0.171 V (Figure 3A, orange), in acetate buffer pH 5 when only three peaks are visible, with the last peak around 1 V not visible (Figure 3A, light green), and in PBS pH 6 when only the first two peaks can be observed (Figure 3B, bordeaux). In a neutral or basic environment, only the first two peaks are visible. As pH increases, the peak around 0.2 V shifts to a more negative potential (Figure 3B).

In all cases, the peak around 0.2 V showed the highest amplitude. The peak amplitude showed a gradual decrease with increasing pH for each electrolyte, with the exception of H_2_SO_4_ 1 M, when the height of the peak was lower than that obtained in H_2_SO_4_ 0.5 M. The best results were obtained in H_2_SO_4_ 0.5 M (Figure 3A, red), which was used as the supporting electrolyte for the further studies.

#### 3.1.3. The Appropriate Electrochemical Technique for PQS Detection

While CV is used to characterize the redox process, a more sensitive electrochemical technique is needed for the quantification of the analyte. Two voltametric methods were tested, SWV and DPV. The advantages of SWV proved to be the rapidity and a more uniform blank, with less background noise. However, in DPV the peak current is higher, 79.78 µA, compared to 38.85 µA generated by SWV (Figure 4) despite using an SR five times higher for SWV to boost the signal. A comparison of the two techniques while maintaining the same electrochemical parameters can be found in Appendix A. Despite the advantages, SWV is less sensitive and DPV was chosen as the most appropriate technique.

The DPV parameters were optimized and the highest current intensity was obtained for an SP of 0.005 V, MA 0.1 V, MT 0.025 s, and an SR of 0.01 Vs^−1^ (data shown in Appendix A).

#### 3.1.4. Influence of the Scan Rate

CV was used to show the influence of the SR on the three anodic peaks of the PQS fingerprint and to determine the process that describes the electrochemical oxidation of the PQS at CNT-SPE. As shown in Figure 5, the current intensity of the oxidation peaks increases with the increase in the SR. For the first and second peaks, the current intensities (Ipa) are proportional to the square root of SR (Figure 5B). The linear regression equations are: Ipa (μA) = 7.929x − 7.8053 (mV^1/2^ s^−1/2^) with a correlation coefficient (R^2^) of 0.989 and Ipa (µA) = 0.4367x − 1.6278 with R^2^ = 0.993, indicating that the oxidation of PQS is controlled by the diffusion of the molecule to the CNT surface. For the last peak, the current is proportional to the SR, but in this case the peak appears to correspond to the oxidation of the adsorbed species. The linear regression equation for this peak is Ipa (µA) = 0.0264x + 0.7159 with R^2^ = 0.9954.

### 3.2. Calibration Curve and Limit of Detection

Different concentrations of a standard solution of PQS (50 nM, 75 nM, 100 nM, 250 nM, 500 nM, 750 nM, 1 µM, 2.5 µM, 5 µM, 7.5 µM, 10 µM, 15 µM, and 20 µM) were tested using the optimized method to analyze the analytical performance of the sensor. The anodic peak around 0.25 V increased with the PQS concentration (Figure 6A). A good linear relationship between the current height and the PQS concentration was observed in the 50 nM–20 µM range (Figure 6B). The linear regression equation is: y (peak height) = 1.2974x (conc.(µM)) − 0.2222, R^2^ = 0.9992.

Equation (2) was used to determine the LOD.
St ≥ Sb + 3 s(2)
where:

St = the signal of the analyte;

Sb = the signal of the blank;

s = the standard deviation of five blank determinations.

Thus, the calculated LOD was 50 nM.

At concentrations lower than 20 μM, an increased accuracy is observed, with very small standard deviations for the three analyses on each concentration tested. At concentrations higher than 20 μM, a larger variation in the obtained values is observed, with higher standard deviations (data not shown). This was also observed previously by Lepine et al. [21] and Oziat et al. [12] and is probably due to the low solubility of the PQS molecule in aqueous solution, which favors the rapid adsorption of PQS on the electrode surface.

Table 3 presents the comparison of the analytical performance of the CNT-SPE sensor with several other electrochemical methods, employing different electrode materials. The sensor developed in this study presents the broadest linear range and the smallest LOD among the voltametric techniques. The low LOD allows the detection of PQS in different biological and microbiological samples of practical relevance. Besides the great analytical performance, the CNT-SPE has several advantages compared to the other methods: easy and low-cost fabrication, single-use, simple and fast sensing procedure, and the possibility to perform decentralized analyses.

### 3.3. Reproducibility Studies

To demonstrate the reproducibility of the developed method, a solution containing 1 µM PQS in H_2_SO_4_ 0.5 M was analyzed on five different CNT-SPE (Figure 7A). The relative standard deviation (RSD%) was 1.4%, indicating a very good reproducibility. A decrease in the peak was observed for successive analyses on the same CNT-SPE (Figure 7B) indicating a possible adsorption of the oxidation products on the electrode surface. These tests indicate that the CNT-SPE is for single use only.

### 3.4. Interference Studies

The selectivity of the sensor was evaluated towards molecules that could be found together with PQS in real samples. The selectivity was tested by measuring the electrochemical signal of PQS in the presence of two antibiotics (GEN and CEF) and an antipyretic drug (APAP), all of them used in the treatment of *P. aeruginosa* infections [24]. The selectivity towards other AIs molecules (C_4_-HSL and 3-O-C_12_-HSL), a *P. aeruginosa* virulence factor (PYO) and biological molecules (GLU, UA) was also evaluated. The DPV signal was recorded in a 5 µM solution containing the interferent alone and in a mixture of the PQS and each interferent and the PQS peak was compared with the peak corresponding to a 5 µM solution of PQS. All the analyses were made in triplicate and the signals in Figure 8 correspond to the average of the signals and the error bars to the relative standard deviations.

The sensor proved to be very selective for PQS detection, with limited influence from the presence of the other AIs molecules (C_4_-HSL and 3-O-C_12_-HSL), CEF, or glucose (Figure 8). GEN is a large molecule, with a tendency to be adsorbed at the surface of the electrode [25] impairing the access of PQS to the electrode, so a smaller peak for PQS is observed in the presence of GEN. The electroactive molecules (APAP, UA, and PYO) have a stronger influence on the peak potential and peak height corresponding to PQS, but still in a small proportion. The anodic peak for PQS does not overlap with the electrochemical signal of these molecules (Figure 9).

The method was optimized in terms of pH, electrolyte, potential window, electrochemical technique for the detection of PQS, but the interference study showed promising results, suggesting that the method could be adapted for the simultaneous detection of PQS and PYO on the CNT-SPE, offering increased selectivity for the diagnosis of *P. aeruginosa* infection.

### 3.5. Real Samples Analysis

#### 3.5.1. Spiked Human Urine, Serum, and Culture Media

The applicability for real sample analysis of the electrochemical method using CNT-SPE for the detection of PQS was tested by analyzing spiked human urine and serum samples and spiked culture media, represented by NB. The real samples analysis required no pretreatment, just a 1:10 dilution with the electrolyte solution. Thus, the real samples were spiked with PQS (10 μM) and then treated with H_2_SO_4_ 0.5 M solution, reaching a final concentration in the analyte of 1 μM. The final solution was analyzed in the optimal conditions using the CNT-SPE. The concentration of PQS in the tested samples was determined using the standard addition method. Very good recoveries were obtained for all three real samples, with limited interference from the complex matrices, proving the utility of the method for the detection of *P. aeruginosa* (Table 4).

#### 3.5.2. *P. aeruginosa* Cultures Analysis

The concentrations of PQS from two *P. aeruginosa* cultures (a standard strain (ATCC 27853) and a clinical isolate) were evaluated at different moments in their growth (after 16, 24, 48, and 72 h) using the developed method. With increasing the incubation time, an increase in the peak corresponding to PQS was observed (Figure 10).

The PQS concentration was compared to the number of colonies (CFU mL^−1^), estimated in parallel (Table 5).

As expected, the number of colonies grew exponentially in time (Figure 11) for both ATCC strain and the clinical isolate. In the standard ATCC strain, PQS concentration had a similar exponential increase to the number of colonies (Figure 11A), but in the clinical isolate, the concentration of PQS grew more rapidly compared to the number of colonies (Figure 11B) showing the increased virulence of this strain and the possibility to detect *P. aeruginosa* in the early stage of infection, ahead of the classical microbial counting method. These results confirm data from the literature that claims that the amount of PQS found in the extracellular medium is correlated with the type and stage of infection [9]. In some cases, where strains produce larger amounts of virulent metabolites, the amount of PQS may exceed the population density (Figure 11).

The results of the analysis of real samples show that CNT-SPE can be used for the evaluation and characterization of *P. aeruginosa* infection, with the PQS concentration correlating with bacterial growth and virulence of the bacterial strain.

## 4. Conclusions

A new electrochemical sensor using commercially available SPE modified with CNTs was developed for the selective and sensitive detection of PQS, a molecule characteristic for the QS of *P. aeruginosa*. The detection method was based on the electrochemical oxidation of PQS. The electrochemical behavior of PQS was determined using different electrode materials (GCE, CPE) and SPEs (BDD-SPE, Pt-SPE, C-SPE, C-SPE modified with AuNPs, OMC-SPE, CNT-SPE, GPH-SPE, and GNP-SPE). The highest sensitivity and reproducibility were obtained using CNT-SPE, DPV as the electrochemical technique and H_2_SO_4_ 0.5 M as the supporting electrolyte. The developed sensor was able to successfully detect PQS in the wide range of 0.05–20 µM, with a very low LOD of 50 nM. The electrochemical sensor proved to be very selective for PQS detection, with limited influence from other AI molecules, antibiotics, paracetamol, and biological compounds. The anodic peak for PQS was not even affected by the presence of other electroactive compounds. The sensor led to very good recoveries when spiked urine and serum samples and spiked microbiological growth media were analyzed. The developed method could successfully determine the concentration of PQS from two *P. aeruginosa* cultures evaluated at different moments in their growth.

## Figures and Tables

**Figure 1 biosensors-12-00638-f001:**
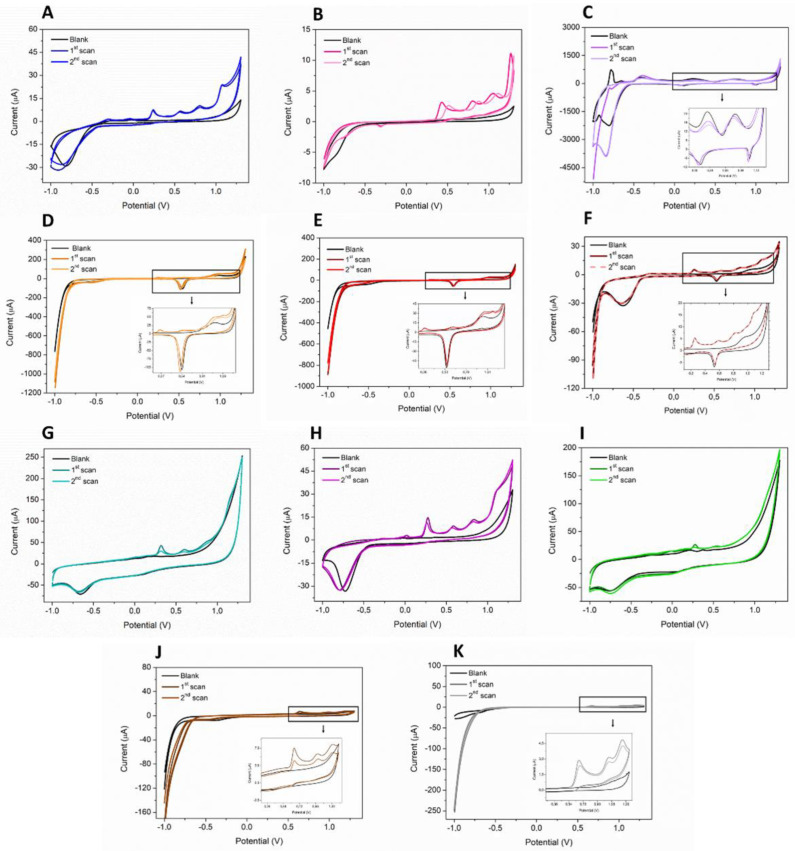
CVs of 50 µM PQS in 0.5 mM H_2_SO_4_ on: (**A**) C-SPE, (**B**) BDD-SPE, (**C**) Pt-SPE, (**D**) C-SPE modified with 35 CV cycles of 5 mM HAuCl_4_, (**E**) C-SPE modified with 35 CV cycles of 1.5 mM HAuCl_4_, (**F**) GNP-SPE, (**G**) OMC-SPE, (**H**) CNT-SPE, (**I**) GPH-SPE, (**J**) GCE, and (**K**) CPE; SR 50 mV s^−1^.

**Figure 2 biosensors-12-00638-f002:**
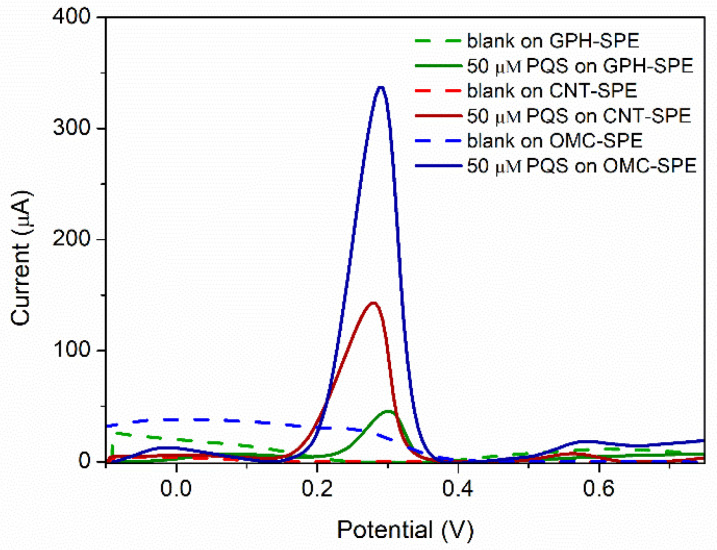
DPV analysis of 50 µM PQS in H_2_SO_4_ 0.5 M on SPEs modified with carbon-based nanomaterials.

**Figure 3 biosensors-12-00638-f003:**
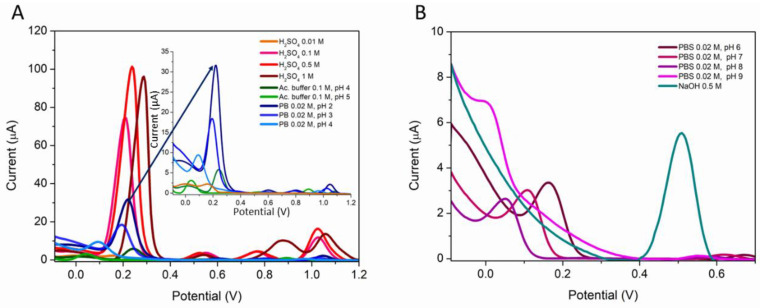
DPV analysis on CNT-SPE of 50 µM PQS prepared in different electrolytes: (**A**) acidic pH and (**B**) neutral and alkaline pH.

**Figure 4 biosensors-12-00638-f004:**
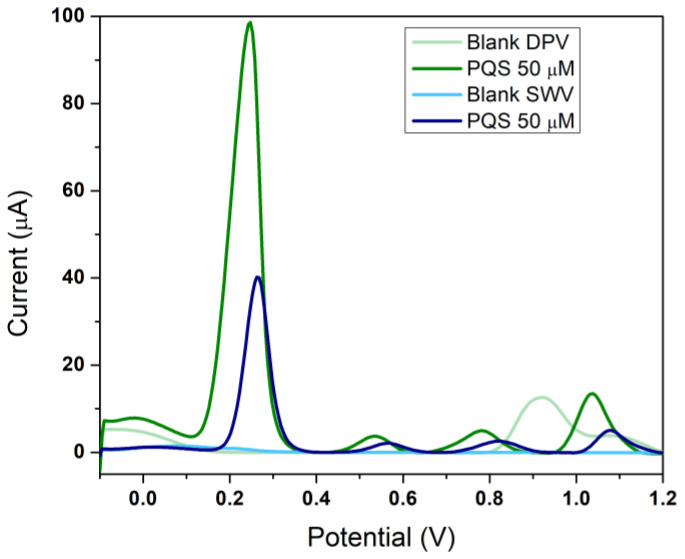
The influence of the electrochemical technique for PQS detection: blank (H_2_SO_4_ 0.5 M) analysis by DPV (**light green**) and SWV (**cyan**); analysis of a 50 µM PQS solution by DPV (**dark green**) and SWV (**dark blue**).

**Figure 5 biosensors-12-00638-f005:**
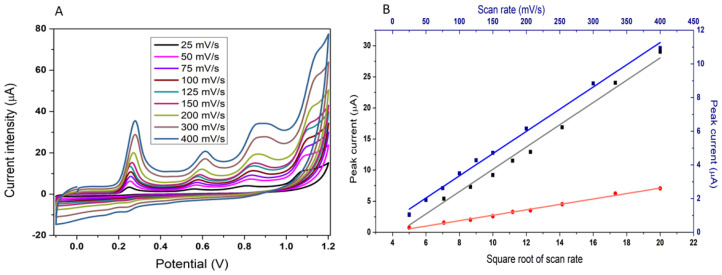
(**A**) The evolution of the three peaks of PQS with the change in SR from a 50 µM solution in H_2_SO_4_, (**B**) the suitable fittings of the peak currents at each SR describing the oxidation process: peak 1 (**black**); peak 2 (**red**); peak 3 (**blue**).

**Figure 6 biosensors-12-00638-f006:**
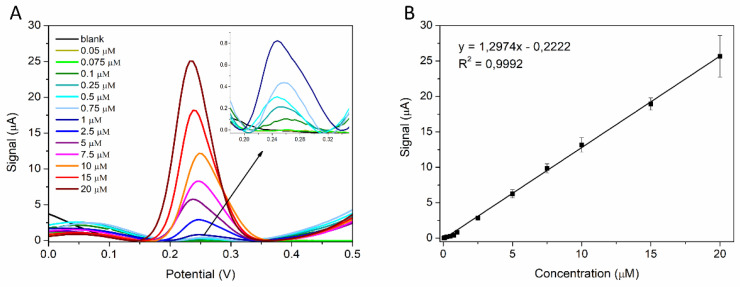
(**A**) The DPV analyses of PQS solutions at different concentrations (50 nM, 75 nM, 100 nM, 250 nM, 500 nM, 750 nM, 1 µM, 2.5 µM, 5 µM, 7.5 µM, 10 µM, 15 µM, and 20 µM); (**B**) linear fitting of the peak height and the concentration of PQS.

**Figure 7 biosensors-12-00638-f007:**
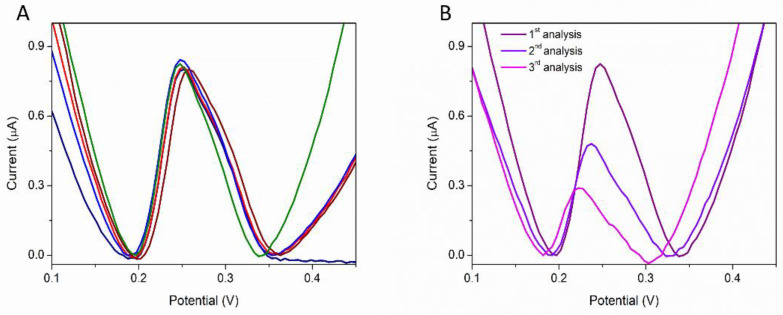
DPV analysis of 1 µM PQS in H_2_SO_4_ 0.5 M: (**A**) on five different CNT-SPE and (**B**) successive analyses on the same CNT-SPE.

**Figure 8 biosensors-12-00638-f008:**
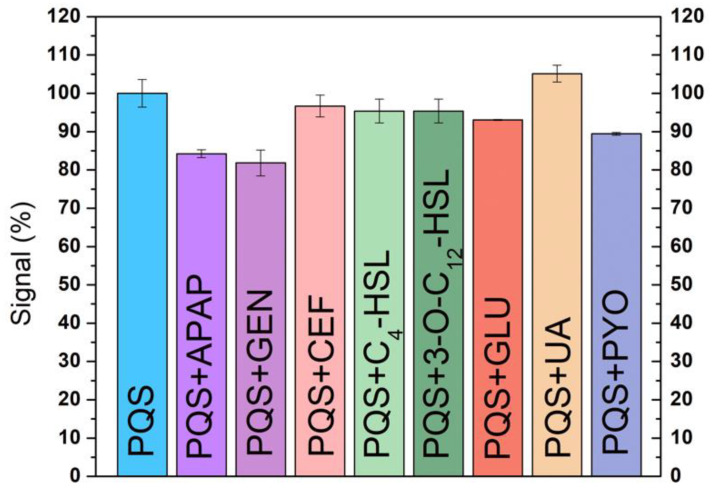
Selectivity of the SPE-CNTs for PQS.

**Figure 9 biosensors-12-00638-f009:**
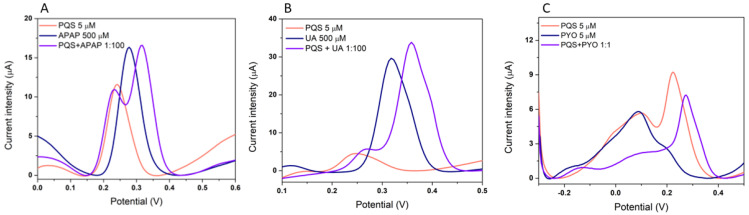
Interference with other electrochemically active substances: (**A**) APAP, (**B**) UA, and (**C**) PYO.

**Figure 10 biosensors-12-00638-f010:**
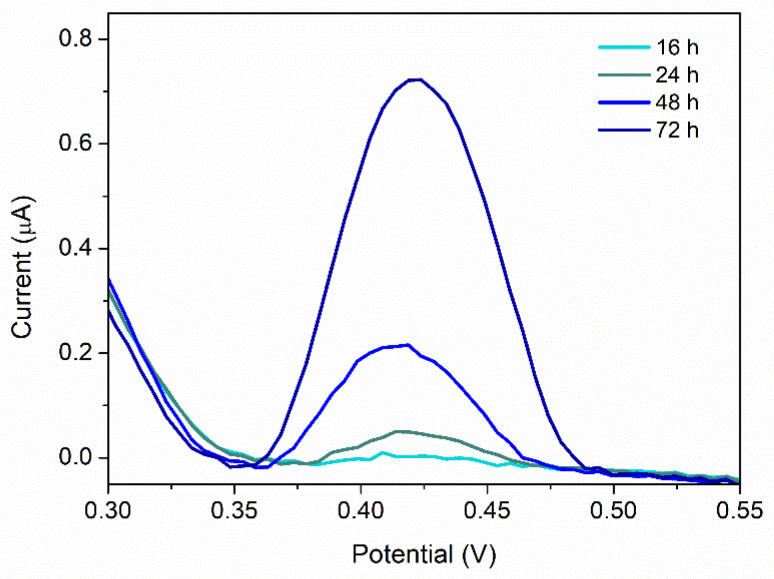
DPV analysis of *P. aeruginosa* cultures (dilution NB:H_2_SO_4_ 1:10) at different moments in their growth (the results for the clinical isolate).

**Figure 11 biosensors-12-00638-f011:**
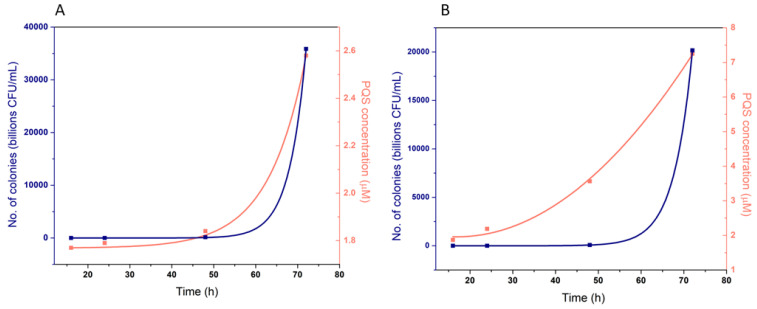
Correlation between the concentration of PQS, determined with the sensor and the number of bacterial cells of *P. aeruginosa* (**A**) ATCC 27853 strain and (**B**) clinical isolate. The number of bacterial cells (**blue**, left O-Y axis) and the concentration of PQS determined with the sensor (**orange**, right O-Y axis) as a function of time (h).

**Table 1 biosensors-12-00638-t001:** The height/position of the peaks obtained by CV of a 50 μM PQS in 0.5 M H_2_SO_4_ on different electrode materials (1st scan).

Electrode	Peak 1		Peak 2		Peak 3		Peak 4	
	Height (µA)	Position (V)	Height (µA)	Position (V)	Height (µA)	Position (V)	Height (µA)	Position (V)
**C-SPE**	5.741	0.240	2.117	0.565	2.638	0.802	6.547	1.070
**BDD-SPE**	2.358	0.430	0.912	0.802	1.091	1.0458	−3.000	−0.304
**Pt-SPE**	-	-	5.500	0.579	-	-	-	-
**C-SPE/AuNPs (35 cycles of 1.5 mM HAuCl_4_)**	4.860	0.269	2.046	0.603	6.856	0.987	1.034	1.099
**C-SPE/AuNPs (35 cycles of 5 mM HAuCl_4_)**	8.724	0.252	3.039	0.565	10.212	0.950	2.640	1.078
**GNP-SPE**	3.608	0.252	0.903	0.582	1.163	0.816	1.556	1.082
**OMC-SPE**	21.640	0.318	6.200	0.600	1.792	0.858	6.240	1.150
**CNT-SPE**	11.913	0.274	3.241	0.582	3.161	0.826	5.646	1.104
**GPH-SPE**	8.301	0.276	1.890	0.525	0.034	0.857	0.697	1.146
GCE	-	-	3.542	0.640	0.742	0.980	0.923	1.202
CPE	-	-	2.206	0.667	0.632	1.034	1.292	1.212

**Table 2 biosensors-12-00638-t002:** The height/position of the peaks obtained by DPV on CNT-SPE of a 50 μM PQS in different electrolytes.

Electrolyte	Peak 1		Peak 2		Peak 3		Peak 4	
	Height (µA)	Position (V)	Height (µA)	Position (V)	Height (µA)	Position (V)	Height (µA)	Position (V)
**H_2_SO_4_ 1 M**	95.877	0.287	2.672	0.544	7.764	0.871	11.091	1.058
**H_2_SO_4_ 0.5 M**	99.677	0.237	3.672	0.524	4.613	0.771	16.494	1.023
**H_2_SO_4_ 0.1 M**	73.367	0.207	3.741	0.549	4.456	0.766	11.983	1.028
**H_2_SO_4_ 0.01 M**	1.275	0.171	-	-	-	-	-	-
**PB 0.02 M, pH 2**	28.532	0.222	0.774	0.600	0.746	0.801	2.293	1.048
**PB 0.02 M, pH 3**	14.823	0.197	0.290	0.590	0.594	0.786	1.268	1.022
**PB 0.02 M, pH 4**	5.780	0.096	0.114	0.564	-	-	0.652	0.967
**Acetate buffer 0.1 M, pH 4**	5.634	0.242	0.125	0.433	0.850	0.871	0.570	1.073
**Acetate buffer 0.1 M, pH 5**	3.135	0.041	0.281	0.529	1.049	0.892	-	-
**PBS 0.02 M, pH 6**	2.315	0.167	0.187	0.670	-	-	-	-
**PBS 0.02 M, pH 7**	2.144	0.111	0.201	0.620	-	-	-	-
**PBS 0.02 M, pH 8**	1.777	0.056	0.114	0.584	-	-	-	-
**PBS 0.02 M, pH 9**	1.512	0.015	0.140	0.554	-	-	-	-
**NaOH 0.5 M**	-	-	5.66	0.509	-	-	-	-

**Table 3 biosensors-12-00638-t003:** Comparison of this sensor with other electrochemical methods previously reported for the detection of PQS.

Electrode	Technique	Linear Range (µM)	LOD (µM)	Interferents	Sample	Ref.
GCE	SWV	5–80	-	-	Standard solution	[15]
GCE/ PEDOT-GCE	SWV	5–70	5	PYO, 2-AA	LB medium with *P. aeruginosa*	[22]
BDDE	DPV	5–50	4.85	PYO, HHQ	Spiked LB broth and CF sputum samples	[11]
BDDE	DPV	2–100	0.25	PYO, HHQ	Spiked sputum samples	[14]
BDDE	Amperometry	-	0.001	HHQ	Culture media with bacteria	[13]
Micro-liquid–liquid interface	ITIES	2–100	1.1	HHQ	Sputum extracts	[23]
CNT-SPE	DPV	0.05–20	0.05	C_4_-HSL, 3-O-C_12_-HSL, PYO, APAP, GEN, CEF, UA, GLU	Spiked urine, serum and culture media; culture media with bacteria	This work

Abbreviations: LB, Lysogeny Broth; PEDOT, Poly(3,4-ethylenedioxythiophene); 2-AA, 2′-aminoacetophenone; ITIES, the electrified interface between two electrolytic solutions.

**Table 4 biosensors-12-00638-t004:** Real sample analysis spiked with 1 μM PQS.

Sample	Spiked PQS Conc. (μM)	Found PQS Conc. (μM)	Recovery (%)
Urine	1	1.18	118 ± 0.05
Serum	1	0.9684	96.84 ± 0.15
NB	1	1.022	102.2 ± 0.10

**Table 5 biosensors-12-00638-t005:** *P. aeruginosa* cultures analysis.

Time	*P. aeruginosa* ATCC 27853		*P. aeruginosa* Clinical Isolate	
	No. of Colonies (CFU mL^−1^)	PQS Conc. (µM)	No. of Colonies (CFU mL^−1^)	PQS Conc. (µM)
16 h	6.35 × 10^7^	1.77 ± 0.01	3.6 × 10^7^	1.87 ± 0.02
24 h	59 × 10^7^	1.79 ± 0.03	32 × 10^7^	2.19 ± 0.02
48 h	14.51 × 10^10^	1.84 ± 0.04	7.85 × 10^10^	3.57 ± 0.01
72 h	3587 × 10^10^	2.58 ± 0.01	2017 × 10^10^	7.25 ± 0.03

## Data Availability

Not applicable.

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
