# Peer review of "Highly Sensitive Detection of PQS Quorum Sensing in Pseudomonas Aeruginosa Using Screen-Printed Electrodes Modified with Nanomaterials"

_biosensors, 2022, doi:10.3390/bios12080638_

Round 1

Reviewer 1 Report

The work describes the fabrication of an electrochemical sensor based on commercial SPEs modified with CNTS for electrochemical detection of pseudomonas quinolone signal (PQS) as a mean for the simple and rapid diagnosis of Pseudomonas aeruginosa infection. DPV technique was used in the measurements and a limit of detection of 50 nM was achieved. The proposed sensor was applied for PQS detection in real samples (urine, serum and microbiological cultures/growth media). The manuscript has scientific interest to many scientists working in the related field and, thus, of interest for Biosensors journal. However, the manuscript should be improved prior to publication considering the following comments:

Specific Comments:

1. Innovative features of the work should be clearly presented in the Introduction, after the state-of-the art. Is it the first time that SPEs are used for detection of Pseudomonas aeruginosa? Is point-of-care (POC) detection of Pseudomonas aeruginosa currently a demand in clinical and diagnosis fields?

2. Why results obtained with GCE and CPE were included in this work (Fig. 1)? It is not clear in manuscript text why they were used. The same for the C-SPEs modified with AuNPs by electrodeposition of a HAuCl4 solution. Why the authors tested this approach since commercial GNP-SPE were also tested.

3. NB is the abbreviation of what type of sample?

4. Fig. 3 can be moved for Supporting Information.

5. In section 3.1.2, what was the medium ionic strength effect on the PQS electrochemical behavior?

6. In section 3.1.3, the authors stated that SWV was less sensitive and DPV and, for that reason, DPV was chosen as the most appropriate technique. However, these two techniques were compared with very different experimental parameters, such as pulse amplitude for example. The authors should compare these two techniques under similar conditions.

In Table 3, I believe that scan rate units are not correct. Table 3 and the corresponding voltammograms for optimization can be moved to Supporting Information.

7. Section 3.1.4 in not well numbered. Moreover, and to improve manuscript clarity and organization, this section could be moved to CV studies shown in 3.1.1.

The reversibility behavior of PQS redox species at CNT-SPEs (Fig. 6) was similar/significantly different from PQS behavior at similar carbon-based surfaces reported in the literature?

8. What was the medium used in section 3.2 to obtain the represented calibration curve? Standard solution? Or real samples?

The method used to determine the LOD is not clear. Eq. 2 only refers to electrochemical signals recorded.

9. Did the authors studied the sensor selectivity against HHQ?

10. In section 3.5.2, the authors showed a large shift of PQS oxidation potential. Why this large shift occurs? Did the authors tried to dilute samples in buffer solution instead od acid to avoid such drifts? 

Author Response

Reviewer 1

The work describes the fabrication of an electrochemical sensor based on commercial SPEs modified with CNTS for electrochemical detection of pseudomonas quinolone signal (PQS) as a mean for the simple and rapid diagnosis of Pseudomonas aeruginosa infection. DPV technique was used in the measurements and a limit of detection of 50 nM was achieved. The proposed sensor was applied for PQS detection in real samples (urine, serum and microbiological cultures/growth media). The manuscript has scientific interest to many scientists working in the related field and, thus, of interest for Biosensors journal. However, the manuscript should be improved prior to publication considering the following comments:

We thank the reviewer for the useful suggestions.

Specific Comments:

  1. Innovative features of the work should be clearly presented in the Introduction, after the state-of-the art. Is it the first time that SPEs are used for detection of Pseudomonas aeruginosa? Is point-of-care (POC) detection of Pseudomonas aeruginosa currently a demand in clinical and diagnosis fields?

We thank the reviewer for this observation. The introduction has been improved integrating the reviewer’s suggestion.

This is the first approach to detect P. aeruginosa through PQS using SPEs. It is not the first approach of detecting Pseudomonas by using SPEs. Our group started a complex study years ago demonstrating the usefulness of using decentralized methods for bacteria detection (10.1021/acs.analchem.8b01915; 10.1016/j.bioelechem.2017.11.014 etc). While the phenomenon of resistance is increasing and P. aeruginosa also possesses the ability to form biofilm, the need for a POC detection is essential. The classical microbial count takes up to 24 hours for certain results, therefore a more rapid, cheaper, and sensitive detection that can be achieved through SPEs is required. A faster detection gives more time for the health professionals to establish a treatment scheme and increase the chances of survival of P. aeruginosa-infected patients.   

  1. Why results obtained with GCE and CPE were included in this work (Fig. 1)? It is not clear in manuscript text why they were used. The same for the C-SPEs modified with AuNPs by electrodeposition of a HAuCl4 solution. Why the authors tested this approach since commercial GNP-SPE were also tested.

We thank the reviewer for this observation. The two electrode types, GCE and CPE, were used in the optimization step to determine the best surface that can detect PQS with the highest sensitivity. GCE and CPE were compared together with several other screen-printed electrodes. Also, GCE has been used in a three-electrode cell for PQS detection in other studies as well. Thus, we wanted to test this surface to compare our results with those previously reported in the literature. The two electrodes (GCE, CPE) were outperformed by CNTs-SPE, which is another advantage of CNT-SPE together with the fact that they are more suitable for miniaturization and point-of-care diagnosis.

The C-SPEs modified with AuNPs by electrodeposition of HAuCl4 were used for the same purpose, as a preliminary testing surface. The homemade AuNPs-SPEs were tested despite the previous results obtained with GNP-SPEs because the morphology of their structure is different and leads to different outcomes when it comes to analytical performance. There are cases when AuNPs-SPEs obtained by electrodeposition give better results and they are preferred as they are cheaper than GNP-SPE.

  1. NB is the abbreviation of what type of sample?

NB stands for nutrient broth, the growth media used in this study to cultivate P. aeruginosa. The abbreviation is explained in section 2.1 Materials. The nutrient broth was tested both as a spiked sample and as real sample inoculated with two P. aeruginosa strains.

  1. Fig. 3 can be moved for Supporting Information.

We thank to the reviewer for this observation. Fig. 3 was moved to SI.

  1. In section 3.1.2, what was the medium ionic strength effect on the PQS electrochemical behavior?

Following the pH studies, we observed that the best signal was obtained in the acidic range. We have chosen H2SO4 as electrolyte, which can support more acidic values. We further studied the influence of the ionic strength on the H2SO4 solution. With the increase in ionic strength, we observed a shift in the oxidative potential to the right up to the H2SO4 1 M solution. The peak current also increased to a maximum of 99.67 µA obtained in H2SO4 0.5 M. A double increase in ionic strength does not lead to a further increase in the signal, therefore H2SO4 0.5 M is the optimal electrolyte that assures the pH and the ionic strength for PQS detection.

  1. In section 3.1.3, the authors stated that SWV was less sensitive and DPV and, for that reason, DPV was chosen as the most appropriate technique. However, these two techniques were compared with very different experimental parameters, such as pulse amplitude for example. The authors should compare these two techniques under similar conditions.

We thank the reviewer for this observation. We have recorded the signal of PQS by SWV using the same parameters as in DPV (potential window -0.1 to 1.3 V, modulation amplitude 0.1 V, scan rate 0.01 V/s). The signal obtained in SWV was lower than expected so we increased the scan rate to 0.05 V/s to boost the signal in SWV and those results were presented in the paper. A comparison between the two electrochemical techniques using the same parameters is presented in the following figure (Figure 1). The signal obtained in DPV is even higher while maintaining the same parameters, showing its advantages over SWV.

Figure 1. The influence of the electrochemical technique for PQS detection: blank (H2SO4 0.5 M) analysis by DPV (light green) and SWV (cyan); analysis of a 50 µM PQS solution by DPV (dark green) and SWV (dark blue)

In Table 3, I believe that scan rate units are not correct. Table 3 and the corresponding voltammograms for optimization can be moved to Supporting Information.

We thank to the reviewer for this remark. The unit of measurement for the scan rate has been modified. As suggested, table 3 and the corresponding voltammograms were to SI.

  1. Section 3.1.4 in not well numbered. Moreover, and to improve manuscript clarity and organization, this section could be moved to CV studies shown in 3.1.1.

We thank to the reviewer for this remark. The section number was corrected.

Although the same electrochemical technique (CV) is used, the two sections present different topics. In section 3.1.1 the influence of the electrode surface on the electrochemical behavior of PQS is presented, while in section 3.1.4 is presented the influence of an electrochemical parameter (scan rate) and is determined the process that describes the electrochemical oxidation of PQS on the CNT-SPE.

The reversibility behavior of PQS redox species at CNT-SPEs (Fig. 6) was similar/significantly different from PQS behavior at similar carbon-based surfaces reported in the literature? 

As this is the first approach of using SPEs for the detection of Pseudomonas aeruginosa through PQS we did not find similar articles reporting the reversibility behavior of PQS redox species at similar carbon-based surfaces. However, GCE has previously been used for PQS detection and we observed that the reversibility behavior of PQS redox species at GCE in our study was like the behavior of PQS at the same platform in previous studies. Also, we observed in our study that the electrochemical behavior of PQS on GCE is like that observed on CNT-SPE, with the difference that in the case of GCE only 3 oxidation peaks can be observed, compared to 4 peaks in the case of CNT-SPE and that the height of these peaks is higher in the case CNT-SPE.

  1. What was the medium used in section 3.2 to obtain the represented calibration curve? Standard solution? Or real samples?

The calibration curve was obtained using standard solutions of PQS obtained by diluting the stock solution with the electrolyte, H2SO4 0.5 M.

The method used to determine the LOD is not clear. Eq. 2 only refers to electrochemical signals recorded.

The equation is formulated as a condition. The LOD is the concentration of the analyte that gives a signal (St) that reaches a value equal or higher to the sum between the signal of the blank solution (Sb) and three-time the standard deviation of five blank determinations. In our case, the signal of the blank was zero (no peak is observed), the standard deviation of five blank determinations was 0.00115 µA, giving a sum of 0.00345 µA, which is inferior to the signal obtained for the 50 nM PQS solution (0.011 µA).

  1. Did the authors studied the sensor selectivity against HHQ?

The selectivity against HHQ was not tested as PQS is the main QS signaling molecule and most of the HHQ is metabolized to PQS. We considered that the hindrance of HHQ would be insignificant to the quantification of PQS.  Also, the selectivity against HHQ was previously tested and it was observed that HHQ did not interfere with PQS. The electrochemical behavior of HHQ is different from that of PQS, showing oxidation peaks at much higher potentials than the potential corresponding to the PQS peak used in our study (around 0.2 V). Thus, in our study we focused on determining the selectivity of PQS against other molecules that can be found in biological fluids.

  1. In section 3.5.2, the authors showed a large shift of PQS oxidation potential. Why this large shift occurs? Did the authors tried to dilute samples in buffer solution instead of acid to avoid such drifts?

The shift in potential observed when analyzing PQS from growth media is generated by the complex matrix of the nutrient broth. The high concentration in components of the media is interacting with the surface of the electrode making it more difficult for the PQS to be oxidized, so the system compensated through a more positive potential. An intermediate dilution in buffer was not performed as we tried to eliminate any extra dilution of the sample and to simplify the work protocol. Moreover, the potential shift does not hinder the quantification of PQS as we obtained good results that correlated well with the number of colonies.

Reviewer 2 Report

The manuscript entitled " Highly selective detection of PQS quorum sensing in Pseudomonas aeruginosa using screen printing electrodes modified with nanomaterials " describes the detection of specific molecule associated with P. aeruginosa infection. Authors had performed extensive investigation to define the optimal experimental conditions such as electrode type, optimal working electrolyte and pH as well as most suitable voltammetric technique. I recommend​ the manuscript for publishing in Biosensors journal after minor changes:

1) All the data presented in the tables' should be provided with standard deviations of the values and with the number of experimental points.​ Please add missed data and explanation for the "*" symbol.

2) The results presented in the literature are someway better than those shown in the work. How can the authors confirm the necessity and advantages of the approach proposed?

3) How can the authors explain the difference in the signals recorded with the electrodes modified with commercial and home-made Au nanoparticles?

4) There are several peaks on Fig. 1 (inset) but only one discussed. Why was another peak excluded from the experiment?

5) It is unclear from Fig. 6, what colour corresponds to what peak (1st, 2nd, 3rd).

6) In Line 333, the authors should check the first equation (v1/2).

7) Background signals on Figs 2 and 3 look absolutely different  as well as the signal changes after the analyte addition. The curves are shifted in the opposite directions on Fig. 2 and 3. Please explain it.

8) Add caption to the inset on Fig. 4A.

9) In Table 2, the authors wrote that results were improved with the pH rising. Why has 1M H2SO4 not been chosen as an optimal media?

10) In Line 232, the authors wrote about "no reduction signal" but they were, also with the Au nanoparticles. Please correct this phrase.

11) In line 229, the authors wrote about " quasi-reversible behavior". Please explain what was meant.

Author Response

Reviewer 2:

The manuscript entitled " Highly selective detection of PQS quorum sensing in Pseudomonas aeruginosa using screen printing electrodes modified with nanomaterials " describes the detection of specific molecule associated with P. aeruginosa infection. Authors had performed extensive investigation to define the optimal experimental conditions such as electrode type, optimal working electrolyte and pH as well as most suitable voltammetric technique. I recommend the manuscript for publishing in Biosensors journal after minor changes:

We thank the reviewer for the useful observations, which improved the quality of our manuscript.

1) All the data presented in the tables' should be provided with standard deviations of the values and with the number of experimental points. Please add missed data and explanation for the "*" symbol.

The data presented in the tables were corrected. The data presented in the tables 1 and 2 corresponds to the representative analysis for each parameter tested (electrode material, electrolyte, respectively) and we feel that adding the standard deviation to the tables would only make the reading of the table more difficult, without bringing more useful information.

The ”*” symbol  does not have any special meaning in this case. This was put to help us in the elaboration of the manuscript and was forgotten by mistake. The symbol has been deleted from the table.

2) The results presented in the literature are someway better than those shown in the work. How can the authors confirm the necessity and advantages of the approach proposed?

In comparison with other studies, the current study uses a different electrochemical platform, a carbon-based screen-printed electrode. While the GCE and BDDE require additional reference and auxiliary electrodes and need a large volume of sample, SPEs have all the electrodes integrated into the same platform and a minimum volume for analysis (50 µL) is required. Moreover, the screen-printed electrodes are disposable after one use and do not pose problems like reconditioning or risk of contamination. The LOD is much lower than in the case of other voltammetric methods. The linear range allows us to detect PQS in a wide number of samples like urine, serum, and culture media. Taking this into account, the SPEs are easier to miniaturize and serve better their purpose for point-of-care diagnosis.

3) How can the authors explain the difference in the signals recorded with the electrodes modified with commercial and home-made Au nanoparticles?

The difference in the signal recorded with the commercial GNP-SPE and the homemade AuNPs/C-SPE is given by the difference in their fabrication process and therefore the difference in the morphology of the surface. The homemade AuNPs-SPEs were obtained by electrodeposition of a HAuCl4 solution while cycling the potential between -0.2 to 1.2 V with a scan rate of 0.1 Vs-1. The properties of the final surface depend on the HAuCl4 concentration, the number of cycles, the scan rate, and the potential window. There is a difference not only between the commercial GNP-SPEs but also between the AuNPs/C-SPEs obtained by different electrodeposition methods (1.5 mM and 5 mM concentration of HAuCl4). Moreover, as the AuNPs-SPEs were obtained, they were used the same day while the GNP-SPEs require preconditioning before analysis due to the oxidation processes happening during storage.

4) There are several peaks on Fig. 1 (inset) but only one discussed. Why was another peak excluded from the experiment?

The first peak was the biggest, thus offering the best sensitivity to the method.

5) It is unclear from Fig. 6, what colour corresponds to what peak (1st, 2nd, 3rd).

We thank the reviewer for the remark. We edited the caption of the graphic so a better correlation between the peaks and the colours can be made.

6) In Line 333, the authors should check the first equation (v1/2).

We thank the reviewer for the observation. The correction to the equation was made.

7) Background signals on Figs 2 and 3 look absolutely different  as well as the signal changes after the analyte addition. The curves are shifted in the opposite directions on Fig. 2 and 3. Please explain it.

The background signal of OMC in the two figures is different because the OMC-SPE surface is not reproducible, so each electrode has its signal in the blank solution. In Figure 3 several concentrations of PQS were tested to see which is the trend in the signal with the increase in the concentration. The peak shown in Figure 2 at 0.1 V does not correspond in fact to the oxidation of PQS but to the surface of the OMC-SPE, this peak decreases while more analyses are performed on the electrode. The peak specific to PQS appears at 0.22 V and it is visible starting with the 5 µM PQS solution. Despite the high signal of PQS obtained on the OMC platform in Figure 1, Figure 2 shows the inability of the platform to detect low concentrations of the analyte and the strong influence of the blank.

8) Add caption to the inset on Fig. 4A.

The caption for the inset of fig. 4A was added.

9) In Table 2, the authors wrote that results were improved with the pH rising. Why has 1M H2SO4 not been chosen as an optimal media?

We thank the reviewer for this remark. We have observed a better defined signal in a lower pH range. However, when testing different concentrations of H2SO4 the best results were obtained with a 0.5 M H2SO4 solution. The height of the peak was lower in H2SO4 1 M than that obtained in H2SO4 0.5 M. This additional explanation was also introduced in the manuscript.

10) In Line 232, the authors wrote about "no reduction signal" but they were, also with the Au nanoparticles. Please correct this phrase.

The cathodic peak observed in the case of the C-SPE platform modified with AuNP corresponds to the electrochemical reduction of AuNP. This peak can also be observed in the blank, indicating the reduction of gold on the surface of the electrode. PQS does not show a peak in reduction except on BDD-SPE, as is also mentioned in the text. For better understanding, the text has been improved with explanations.

11) In line 229, the authors wrote about " quasi-reversible behavior". Please explain what was meant.

We thank the reviewer for this observation. The manuscript was modified to say „irreversible behavior”.
